# The Expression and Function of the Small Nonstructural Proteins of Adeno-Associated Viruses (AAVs)

**DOI:** 10.3390/v16081215

**Published:** 2024-07-29

**Authors:** Cagla Aksu Kuz, Shane McFarlin, Jianming Qiu

**Affiliations:** Department of Microbiology, Molecular Genetics and Immunology, University of Kansas Medical Center, Kansas City, KS 66160, USA; caksukuz@kumc.edu (C.A.K.); mmcfarlin2@kumc.edu (S.M.)

**Keywords:** AAV, gene expression, assembly-activating protein (AAP), membrane-associated accessory protein (MAAP), capsid assembly, virus egress

## Abstract

Adeno-associated viruses (AAVs) are small, non-enveloped viruses that package a single-stranded (ss)DNA genome of 4.7 kilobases (kb) within their T = 1 icosahedral capsid. AAVs are replication-deficient viruses that require a helper virus to complete their life cycle. Recombinant (r)AAVs have been utilized as gene delivery vectors for decades in gene therapy applications. So far, six rAAV-based gene medicines have been approved by the US FDA. The 4.7 kb ssDNA genome of AAV encodes nine proteins, including three viral structural/capsid proteins, VP1, VP2, and VP3; four large nonstructural proteins (replication-related proteins), Rep78/68 and Rep52/40; and two small nonstructural proteins. The two nonstructured proteins are viral accessory proteins, namely the assembly associated protein (AAP) and membrane-associated accessory protein (MAAP). Although the accessory proteins are conserved within AAV serotypes, their functions are largely obscure. In this review, we focus on the expression strategy and functional properties of the small nonstructural proteins of AAVs.

## 1. Introduction

Adeno-associated virus (AAV) was first discovered as “DNA-containing particles” in a preparation of simian adenovirus (Ad) in 1965 [1,2]. AAVs, belonging to the genus *Dependoparvovirus* of the family *Parvoviridae*, are a group of non-pathogenic viruses [3]. They possess a T = 1 icosahedral shell of ~25 nm in diameter that packages a single-stranded (ss)DNA genome of ~4.7 kilobases (kb). This genome includes two identical inverted repeats (ITRs) of 145 nucleotides (nts) in length. The species *Dependoparvovirus primate 1* in the genus *Dependoparvovirus* consists of 12 natural AAVs (AAV1–4 and AAV6–13), and the most distantly related AAV5, bovine AAV, and caprine AAV are classified under the species *Dependoparvovirus mammalian 1* [3]. AAV is a naturally defective virus that requires a helper virus either from the families of *Adenoviridae* (e.g., adenovirus 5, Ad5) or *Herpesviridae* (e.g., herpes simplex virus 1, HSV1) to facilitate its viral replication [4,5]. Strikingly, human bocavirus 1 (HBoV1), a member of the *Bocaparvovirus* genus of the *Parvoviridae* family [6,7], is able to provide a helper function to AAV2 for a productive infection [8].

Utilization of recombinant (r)AAV as a gene delivery vector in mammalian cells was initially established as early as 1984 [9]. AAV capsids are engineerable and have a broad tropism to a variety of cells or tissues; therefore, they can be vectorized for gene delivery applications [10,11]. AAV has been extensively studied in molecular virology, vector production, capsid tropism, and efficacy in gene delivery for over 50 years. So far, six rAAV-based gene therapy medicines—Luxturna, Zolgensma, Elevidys, Roctavian, Hemgenix and Beqvez—have been approved by the US FDA for the treatments of Leber congenital amaurosis, spinal muscular atrophy, Duchenne muscular dystrophy, hemophilia A and B, respectively [12,13,14,15,16,17]. Moreover, as of May 2024, over 350 gene therapy clinical trials using rAAV vectors are ongoing worldwide (Clinicaltrials.gov and Wiley Database on Gene Therapy Clinical Trials Worldwide).

rAAV can deliver transgenes to (transduce) host cells of both dividing and nondividing cells for long-term gene expression. Initially, studies on AAVs have been focused on elucidating the biology of the viruses towards manipulation of rAAV vector production for utilization in therapeutic applications [4]. To meet the high and urgent demands in gene delivery applications, research has evolved to capsid engineering for AAV variants that exhibit a high tropism to targeted organs, tissues, or cells [10,18]. While scientists have achieved successful progress in the utilization of rAAV in gene therapy applications, the field has encountered multiple challenges and limitations, ranging from the limitation in the sizes of the packaged DNA, the transduction efficiency, the yield and purity of the vector, and the vector-induced immune response [19].

## 2. AAV Life Cycle

AAV infection is initiated with its attachment to host cells through glycans and a commonly used proteinaceous receptor known as KIAA0319L (also referred to as AAV receptor, AAVR) [20,21], or alternate receptors [22]. While heparan sulfate on cell surfaces is used for AAV2, AAV3, AAV6, and AAV13 attachment, sialic acids are used for AAV1, AAV5, and AAV6, and galactose is used for AAV9 [23,24,25,26,27,28,29]. After attachment and receptor binding, AAV is internalized via several endocytic pathways: clathrin-mediated endocytosis, clathrin-independent carriers/GPI-anchored protein enriched early endosomal compartments (CLIC/GEEC), or macropinocytosis [30,31,32,33] (Figure 1). After internalization, AAV is endocytosed into various endosomes/vesicles; it is then intracellularly trafficked to the trans-Golgi network (TGN)/Golgi apparatus, and likely to the endoplasmic reticulum (ER), and escapes the vesicle membrane, which is mediated by phospholipase A2 (PLA_2_), located on the N-terminus of the unique region of VP1 (VP1u) [34,35,36,37,38] (Figure 1). However, the functions of AAVR and other AAV entry factors, such as G protein-coupled receptor 108 (GPR108) and transmembrane 9 superfamily member 2 (TM9SF2) [39,40], during AAV endocytosis and intracellularly trafficking are currently not fully understood [41]. The intracellular virus reaches the nuclear membrane for nuclear entry through nuclear pore complexes (NPCs) [42]. At the nucleus, AAV is uncoated and releases its ssDNA genome, which is converted to a transcriptional dsDNA genome, followed by the expression of Rep proteins (Figure 1). The Rep proteins are essential for viral transcription activation and viral DNA replication [43,44]. Followed by expression, the viral capsid proteins (VP1, VP2, and VP3) are transported to the nucleus to reassemble empty capsids. The empty capsids then capture the ssDNA genome generated through strand-displacement during viral dsDNA replication (Figure 1). The preassembled capsids will then package either negative or positive ssDNA at a 50% frequency [45]. The virus is finally maturated in the nucleolus and is transported to the cytoplasm for cellular egress [46]. 

## 3. AAV Genome and ITR

As the representative serotype of the genus, the AAV2 genome, flanked by two ITRs, comprises 2 open reading frames (ORFs), with one intron located in between, and one polyadenylation signal at the 3′ terminus [4,49]. The AAV2 ITR is palindromic sequences of 145 nts [50,51,52], forming the characteristic shape of DNA-termini in parvoviruses, the hairpin structure (Figure 2A). The palindromic arms include A-A’, B-B’, and C-C’, which form a T-shape. The arrangement of the B-B’ and C-C’ sequences determines the ITR’s orientation, which can be either “flip” or “flop”. In the flip configuration, the B-B’ palindrome is closer to the 3′ end, while in the flop configuration, the C-C’ palindrome is closer to the 3′ end. The AAV genome can be flanked by two flip-oriented domains, two flop-oriented domains, or one ITR of each orientation. On the other hand, although the AAV5 ITR shares the same features of the palindromic arms, it is longer with a sequence of 182 nts (Figure 2B). This secondary structure functions as the initiator of viral DNA replication via 3′-hydroxyl group retained as a free 3′ end following the self-folding of the palindromic-125 nts [53]. This DNA arrangement can be re-organized by replacing rep and cap genes with a transgene in rAAV production. In this way, a rAAV2 genome contains the only cis essentials, ITRs, of the virus DNA replication and packaging, which can be replicated by providing Rep proteins, and helper virus functions [9]. During replication of the viral DNA genome (as dsDNA), ITRs are bound by two large Rep proteins (Rep78/68) at the Rep-binding element (RBE). The Rep78/68 exhibit helicase functions to unwind the double helix of the terminal resolution site (TRS) and nick the TRS in a strand-specific manner [54,55] (Figure 1). This linearization of the genome ends leads to replication initiation by acting as the origin of replication (*Ori*) [56]. In addition, both AAV2 and AAV5 ITRs can function as a transcription initiator (Inr) that transcribes AAV RNA [57,58].

## 4. AAV Gene Expression

### 4.1. Rep Gene Expression

AAV2 *Rep* ORF encodes four replication/packaging-related proteins (Rep78, Rep68, Rep52 and Rep40) transcribed by two promoters: P5 and P19, respectively (Figure 3A). The P5 promoter is activated by the Ad gene *E1A* [59], but in the presence of Ad, it was repressed by AAV2 Rep78/68 [60]. Rep78 is translated from an unspliced transcript, whereas Rep68, including Rep68 minor (Rep68m) and Rep68 Major (Reg68M), is encoded by D-A1 and D-A2 spliced mRNAs, respectively (Figure 3A). Rep78/68 proteins convey endonuclease, helicase, and ATPase functions during the viral genome replication [61]. The binding of Rep78/68 to the RBE at the ITR is required for linearization of hairpin structures to allow replication of the ITR [55,62,63,64] (Figure 1). Rep52/40 proteins are encoded from the unspliced and spliced mRNAs, respectively, that are transcribed from the P19 promoter [65]. Activation of the P19 promoter relies on the P5 promoter or the ITR and Rep78/68 [66]. Rep52/40 proteins promote the encapsidation of newly synthesized viral ssDNA genomes into the preassembled empty capsids [67].

The AAV5 *Rep* gene is transcribed by two promoters—P7 and P19, respectively—and polyadenylated at the proximal pA site, (pA)p, located in the middle of the intron, and therefore only Rep78 and Rep40 are encoded [58,68,69] (Figure 3B). Like the AAV2 Rep proteins, AAV5 Rep78 and Rep40 possess functions for viral mRNA transcription, DNA replication, and genome packaging [58,65].

**Figure 3 viruses-16-01215-f003:**
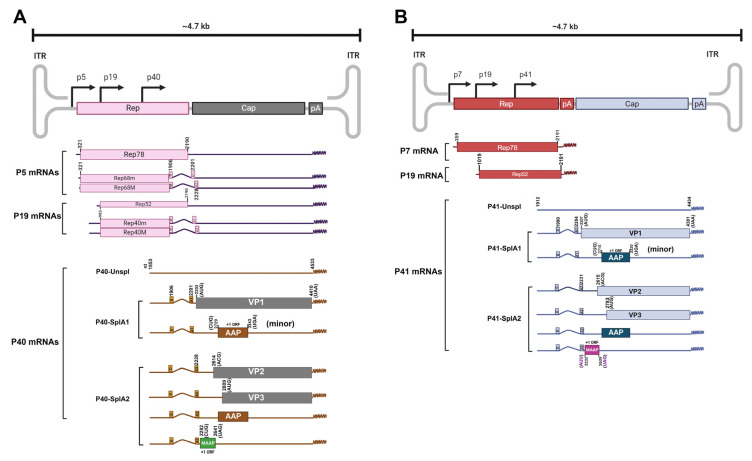
The genetic maps of AAV2 and AAV5. Adapted from [58,65,70]. The ~4.7 kb ssDNA genomes of AAV2 (GenBank accession no. AF043303) (**A**) and AAV5 (AF085716) (**B**) are depicted with two ORFs (*Rep* and *Cap*); one intron that is spliced at one donor (D); two alternative acceptors, A1 as the minor and A2 the major sites, one (AAV2) or two (AAV5) polyadenylation sites (pA); and two ITRs. Three promoters, P5 (AAV2)/P7 (AAV5), P19, and P40, transcribe three sets of AAV mRNAs, P5/P7 mRNAs, P19 mRNAs, and P40/P41 mRNAs, respectively. For AAV2 (**A**), P5 mRNAs encode Rep78, Rep68m, and Rep68M; and P19 mRNAs encode Rep52, Rep40m, and Rep40M. For AAV5 (**B**), P7 mRNA encodes only Rep78, and P19 mRNA encodes Rep40. P40 (AAV2)/P41 (AAV5) mRNAs include P40 unspliced mRNA and alternatively spliced mRNAs at D and A1 or A2, designated as P40/P41-SplA1 and P40/P41-SplA2 mRNA, respectively. SplA1 mRNA encodes VP1, and SplA2 mRNA encodes MAAP, VP2, AAP, and VP3 at the initiation codons as indicated. MAAP and AAP are translated from the +1 frame-shifted ORF. Created in Biorender.com (accessed on 25 July 2024).

### 4.2. Cap Gene Expression

The AAV2 *Cap* gene is transcribed by the P40 promoter, which is completely inactive by itself. The P40 promoter is transactivated by Rep78/68 through binding to RBEs at either the ITR or the P5 promoter [60,71], and splicing of the P40 pre-mRNA is facilitated by Rep [72]. However, the AAV5 P41 promoter is active per se without having AAV5 Rep78 to activate [58,68,69]. Remarkably, the P40/P41 *Cap* gene expresses not only 3 structural proteins, VP1–3, but also two small nonstructural proteins, AAP and MAAP [70,73,74]. During AAV2 or AAV5 infection, the P40/P41 pre-mRNA is alternatively spliced to generate two mRNAs that translate to the viral structural proteins in-frame and nonstructural proteins in shifted-frames using either canonical (conventional) or non-canonical (alternate) start codons (Figure 3) [70]. 

#### 4.2.1. Structural Proteins

VP1, VP2 and VP3 of both AAV2 and AAV5 are viral structural (capsid) proteins that are expressed from overlapping ORFs via conventional or alternate initiation codons (Figure 3). VP1 is expressed from a minor spliced mRNA at the A1 site, whereas VP2 and VP3 are expressed from the major P40/P41 transcript spliced at the A2 site. VP1 is the larger viral protein (~85 kDa) with VP1u located at the N-terminus of the protein, which separates it from other overlapping proteins. VP2 (~70 kDa) and VP3 (~60 kDa) aa sequences are entirely contained within the VP1 sequence at the C-terminus [75]. Although overlapped VP3 within VP1 and VP2 is critical for capsid assembly, efficient AAV infection or rAAV transduction necessitates one PLA2 domain on VP1u and 3 nuclear localization signals (NLS), which are located within VP1u (1 × NLS) and the VP2-unique region (VP2u; 2 × NLS) diverse from VP3 [76,77,78,79]. During infection, a conformational change in the capsid is necessary, resulting in the protrusion of the N-termini of VP1u, leading to endosomal escape via PLA2 activity, and of the VP2u out of the intact capsid, routing the capsid to the nucleus by interacting with the NLSs [37,80].

The ~25 nm T = 1 icosahedral capsid is a combination of 60 viral proteins at a molar ratio of 1:1:10 for VP1:VP2:VP3 [81,82,83], where newly synthesized negative or positive-sense ssDNA genomes (at a ratio of 1:1) are packaged into [84]. The capsid is assembled in the nucleoli followed by the VP1u and VP2u NLS-mediated input of the cytoplasmic trimerized structural proteins [85,86,87]. 

#### 4.2.2. Accessory Proteins

Besides the large nonstructural proteins, Rep78/68 and Rep52/40, expressed during AAV infection, two small nonstructural proteins, AAP and MAAP, have been identified as accessory proteins. AAP and MAAP are expressed from the same, albeit +1 shifted-ORF present in the cap gene during AAV infection (Figure 3) [70,73,74]. Both AAP and MAAP translations are initiated at non-canonical start codons, CUG for AAV2 and AAV5 AAP (AAP2/5) and AAV2 MAAP (MAAP2) and AUU for AAV5 MAAP (MAAP5), respectively (Figure 3) [70]. 

##### AAP

AAP was first identified in 2010 as a new ORF of the *Cap* gene in the AAV2 genome [73]. AAP is ~200 aa in length and has been detected at a size of ~20–25 kDa depending on the AAV serotypes (Figure 4). The investigation commenced with a disagreement among studies regarding the occurrence of AAV capsid assembly. Some studies reported that capsid assembly takes place only when certain structural proteins are expressed, while others suggested the involvement of VP1–3 [88,89,90,91]. The independent expression of VP3 alone was found to be inefficient in forming a capsid in certain studies. Conversely, some researchers were able to observe the formation of capsids containing only VP3 when co-expressed with VP2 and/or VP1. The success of one approach in forming a capsid comprising only VP3 can be attributed to the inclusion of an intact construct coding for the capsid gene. Additionally, this construct contained mutations at the initiation codons of VP1 and VP2 that effectively prevented their expressions [90,91]. Expression of VP3 from a construct containing only the VP3-coding sequence, achieved by deleting the upstream sequences, proved unsuccessful in forming a capsid [88,89]. This inconsistency led to a study in which researchers generated many constructs which gradually included various sequences in different lengths extended upstream of VP3. Only one construct that comprises only the VP3-coding region starting at nt 2809 in the AAV2 genome failed to result in capsid formation [73]. This suggested that an unknown factor was involved in capsid formation and resulted in the identification of the novel AAV nonstructural protein, AAP [73]. 

Unlike the VP ORFs where structural proteins are produced, AAP is identified as a product of an alternate ORF (+1 shifted). This novel ORF is translated from a non-canonical start codon, CUG, located after the 175th amino acid of AAV2 VP1 [70,92], and a CUG after the 169th aa of the AAV5 VP1 (Figure 3). AAP is a unique protein of the genus as it shows no significant homology with other proteins in the National Center for Biotechnology Information (NCBI) database [92]. It is expressed in all naturally occurring AAV serotypes 1–13, and shares a high degree of homology (>80%) (Figure 4). AAP is structurally predicted to contain hydrophobic regions, T/S (threonine/serine-rich) regions, basic amino acid-rich regions, which potentially interacts with host proteins and viral proteins (Figure 4).

##### MAAP

MAAP is structurally predicted to contain hydrophobic regions, T/S regions, basic amino acid-rich regions (BR1 and BR2) with a cationic amphipathic C-terminal domain (α2 helix) which is a potential element for membrane binding (Figure 5), and shares a high homology of >80% among these naturally occurring AAVs [93,94]. Overexpression of various MAAPs in HEK293 cells by transfection showed similar subcellular localization in the cytoplasm and more strikingly at the cellular membrane [74,94,95]. However, in addition to the periplasmic association, MAAP2 (AAV2 MAAP) and MAAP5 (AAV5 MAAP) appeared to localize in the nucleus during wild-type (wt)AAV infection, and the nuclear localization of MAAP5 is much more abundant [70]. The presence of BR1 and BR2 at the C-terminus of MAAP may function as an NLS and lead to nuclear localization of MAAP [95].

Transcriptional analysis revealed that both P40/P41-driven D-A2 spliced mRNAs express MAAP2 and MAAP5 [70], suggesting that the AAV D-A2 mRNA is a quadcistronic mRNA transcript, expressing four proteins, MAAP, VP2, AAP2, and VP3, in order (Figure 3). It is a unique feature among eukaryotic gene translation with the most known cistrons encoded in eukaryotic cells. In contrast to MAAP2, MAAP5 utilizes a different non-canonical codon, AUU, at nt 2232 immediately after the A2 acceptor, for initiation of translation, resulting in the expression of MAAP5 at 17 aa longer than MAAP2 [70], which may result in the abundant expression of MAAP5 in the nucleus. 

#### 4.2.3. Other Potential Small ORFs

Another AAV gene, *X*, transcribed from a promoter, P81, at the position 81 unit of the AAV2 genome, was first identified in 1999 [96]; it is located at the far end of the AAV2 *Cap* gene. It was thought to code an *X* ORF at 3929–4394 and express an X-protein of 155 aa. 

## 5. Functions of AAV Small Nonstructural Proteins

### 5.1. AAP

AAP exhibits high conservation across all natural AAV serotypes (AAV1–13), and bioinformatic analysis has revealed the presence of a homologous protein produced in all AAVs (Figure 4). AAP was initially reported to play a role in the nuclear/nucleolar translocation of VP3 for AAV capsid assembly and maturation [73,92]; however, the necessity of AAP in capsid assembly is serotype-dependent [97,98,99]. While the capsid formation or vector production of AAV1–3, AAV6–9, and AAVrh10 is AAP-dependent, AAP4, AAP5, AAP11, and AAP12 are dispensable for AAV capsid formation or vector production.

VP3-only capsid formation of AAV5 is not activated by heterologous AAPs, i.e., AAP1 or AAP2, whereas AAV5 VP3-only capsids are formed from the in trans expression of AAP of AAV5, AAP5. However, the AAV5 VP3-only assembly is 2-log less efficient than the control, which is readily expressing the AAV2 *Rep* gene and complete AAV5 *Cap* gene [92]. In addition, although the promotion of heterologous capsid assembly is shown to be possible when different AAPs are employed to trans-complement the function, VP3-only capsid formation of different serotypes is weakly stimulated by AAP5 [98]. AAV4, AAV11, AAV12, and rAAVrh32.33 are the serotypes that can form a VP3-only capsid independent of AAP function [98]. AAV4 is both genetically and structurally close to AAV11, AAV12, and AAVrh32.33 in a unique AAV4 Clade [100,101]. These findings suggested that AAP is not required for forming VP3-only capsids in these distantly related AAV serotypes in AAV4-like and AAV5-like Clades, whereas the capsid formation of the AAVs in Clades A-F [100,101] is AAP-dependent. 

AAP increased capsid protein stability and VP-VP interactions by forming oligomers and changing the conformation of the unassembled VP proteins [73,99]. The AAP domain was functionally dissected (Figure 4) [102]. The T/S region can be replaced by other sequences to engineer AAP without any functional defects, whereas deletion of the HR (hydrophobic region), CC (conserved core), and PRR (proline-rich region) domain either individually or together resulted in a loss of AAP function in stimulating capsid assembly. Also, HR and CC domains, located in the N-terminus of AAP, were revealed to have a function in VP stability and interaction. In parallel, the replacement of the T/S region by different oligomerization domains showed an increase in AAP stability and vector yield slightly. Notably, the BR (basic-rich) region can be replaced by a different NLS with no significant functional change to AAP [102], indicating that it is a bona fide NLS. Notably, in contrast to the nucleolar enrichment of assembled AAV1, 2, 3, and 7 capsids, assembled AAV8 and AAV9 capsids are excluded from the nucleolus, as well as the AAP8 and AAP9, which are similar to that of the AAP-independent AAV5, arguing the role of AAP in the nucleolar shuttling of the structural proteins in AAV capsid assembly [98]. 

In summary, AAP is clarified as a component with functions in the stability of the structural proteins and assembly of AAV capsids either comprising only VP3 or including VP1 and VP2 in AAV serotypes of Clades A-F, but not AAV4 and AAV5-like Clades, through an underlying mechanism that is currently unknown. Nevertheless, it is evident that AAP’s function in capsid assembly is not universally applicable to all AAV serotypes, and its necessity is not absolute. However, the supportive role of AAP in capsid assembly in specific AAVs, along with its potential to enhance packaging efficiency and recombinant vector production, should not be overlooked during vector manufacturing. 

### 5.2. MAAP

Although it was first postulated as a protein that limits virus production via competitive exclusion [74], MAAP is also attributed to the function in AAV egress [94], to viral genome packaging, and to a decrease in the stability of structural proteins [95]. On the other hand, MAAP ablation increased overall virus titer and viral DNA replication [70,103].

MAAP has been shown to promote the cellular egress of some serotypes of rAAV vector at different levels [94]. The recovery of produced rAAV8 from the media of the HEK293 cells transfected with an AAV2 *Rep2* and AAV8 *Cap8* expressing plasmid pRep2Cap8 was ~10-fold more than that from the MAAP8-knockout pRep2Cap8^ΔMAAP^-transduced cells. Interestingly, ~70% of the total produced vectors was recovered from the media, whereas the yield was less than 10% when MAAP8 expression was ablated. Consequently, rAAV8 had less vector recovery from the media in the absence of MAAP8 (<10%), although the transduction efficiencies of the rAAV8 produced with and without MAAP expression showed no difference. However, the absence of MAAP during the production of rAAV1, 2, and 9 resulted in no significant decrease in cell-released vectors, whereas rAAV1 and rAAV9 showed an increase in total vector yield. The trans-complementation with MAAP8 during the production of rAAV1 and rAAV2 failed to rescue the function of the cognate MAAP, and yet it increased the recovery of vectors from the media by 2-fold of that in rAAV9 production [94]. In another study, there was a consistent significant decrease in vector cellular release in rAAV2 production observed when MAAP2 was ablated by introducing an early stop codon [103].

During wtAAV infection, both MAAP2 and MAAP5 increase total AAV production (by ~2-fold) and facilitate virus egress out of the infected cells. AAV2^ΔMAAP2^ and AAV5^ΔMAAP5^, in which the *MAAP* ORF was early terminated, showed a significant increase in the production of progeny virions in the cell lysate and total yield compared to infections of wtAAV2 and wtAAV5 [70]. Strikingly, AAV2^ΔMAAP2^ infection exhibited more than 50-fold less virus recovery from the media, whereas released progeny virions from AAV5^ΔMAAP5^-infected cells decreased by 4.9-fold. Importantly, the MAAP function in infected cells was successfully rescued by complementing a cognate MAAP in AAV2 ^ΔMAAP2^- and AAV5^ΔMAAP5^-infected cells [70]. However, in another study, MAAP ablation via either mutation of the start codon or the introduction of an early stop codon resulted in a reduced titer of produced virus from transfection of infectious AAV2 clones in the presence of Ad5 [95]. Notably, packaging of contaminating DNA was increased in the absence of MAAP, whereas, the in trans expression of MAAP, it rescued the function, and the same mutants displayed a decrease in AAV genome packaging compared to the wtAAV2 [95]. 

Mechanistically, the AAV releases outside the cell are partially promoted by MAAP through AAV–extracellular vesicle (EV) association [94,103]. Isolated EVs from the media of transfected cells producing rAAV8 in the presence or absence of MAAP demonstrated that AAV and MAAP-associated EVs were co-eluted in EV-containing cellular fractions. Furthermore, MAAP was found to be accumulated in the same CD63-positive EV fractions as rAAV8 virions, suggesting a possible association of AAV capsid with CD63-positive EVs [94]. However, MAAP-ablated AAV2 production showed no significant difference in the release of virus-associated EVs than during wtAAV2 production. Yet, MAAP2-mutated AAV2 production demonstrated a decrease in the detection of the viral genome and intact capsid found in the isolated EVs compared to that from wtAAV2, substantiating the role of MAAP in virus egress [103].

MAAP was further studied to improve vector production by generating new MAAP variants through directed evolution, which has resulted in two MAAP2 variants that showed an increase in transgene packaging into the AAV2 capsid, as well as an increase in overall AAV2 titer with a high full/empty capsid ratio, whereas only one of the variants demonstrated higher infectivity. Additionally, one variant enabled a significant increase in packaging transgene into either the AAV6 or AAV9 capsid over the packaging in cells where a cognate MAAP was expressed [103].

Non-enveloped lytic virus egresses through active transports other than through the major lytic pathway [104,105]. For lytic parvoviruses, progeny virions are actively transported from the nucleus to the plasma membrane through vesicles in a gelsolin-dependent manner [106]. For minute virus of mice (MVM), a member of the genus *Protoparvovirus* of the *Parvoviridae* family, progeny particles become engulfed into coat protein complex II (COPII)–vesicles in the ER and are transported through the Golgi to the plasma membrane [107] (Figure 6). In MVM, the large nonstructural viral protein NS1 is involved in virus spread to the cytoplasm [106,108], which may be due to the lack of expression of a MAAP-like protein during MVM infection. We speculate that MAAP is involved in the transportation from the ER to the Golgi, which may be mediated by the COPII complex (Figure 6). Notably, multiple studies have demonstrated that a significant fraction of rAAV associated with EVs are released into the cell culture medium during rAAV vector production [109,110,111,112,113,114]. MAAP plays a role in promoting EV-mediated AAV egress from cells [94]. As some of the MAAP localized in the nucleus, we hypothesize that MAAP interacts with the host proteins involved in nuclear export, e.g., exportins [115], components of COPII-vesicles [107], and EV-associated proteins [116], which will escort AAV capsids across the nuclear membrane and cytoplasm in early or late endosomes that generate EVs that are released from the surface membrane (Figure 6).

In conclusion, MAAP is a novel protein expressed from an alternate reading frame (+1-shifted) with variable translation initiations among serotypes. Although it is suggested to facilitate AAV egress, MAAP plays a role in genome packaging, in which the underlying mechanisms remain elusive. In addition, the functional analysis of MAAP in rAAV production broadens our knowledge of AAV egress, which is poorly understood compared to virus entry. MAAP has the potential to be engineered for the improvement of rAAV vector production and can be utilized as a factor for loading cargo in EVs. The identification of host factors interacting with MAAP holds promise for shedding light on the precise mechanism governing MAAP’s role in the AAV life cycle.

### 5.3. X Protein

The function of the X protein has been shown by the same group who discovered it [117]. Lack of *X* gene expression in a wtAAV2 clone resulted in lower DNA replication levels and production of AAV2 virus in HEK293 cells in the presence of Ad5. Additionally, *X* gene expression has also been shown to increase wtAAV2 autonomous replication in the organotypic epithelial raft culture system (skin rafts) [118]. Mutating the *X* gene in an AAV2 packaging plasmid yielded approximately a ~33% reduction in rAAV production [117]. Considering that the *X* ORF is found in at least one member of every AAV clade [117], the function of the X protein in AAV DNA replication needs to be further assessed in other AAV serotypes.

## 6. Conclusions

Despite AAV’s compact ssDNA genome of 4.7 kb, housing only two genes, its extensive study since its initial discovery continues to unveil novel protein expressions during infection. These discoveries hold the potential for engineering advancements, enhancing its applications and improving vector-manufacturing processes. The identification of small nonstructural proteins, AAP and MAAP, occurred 45 and 54 years, respectively, after the initial discovery of AAV. Both AAP and MAAP function in a serotype-dependent manner. AAP studies provide evidence of its function in capsid assembly, whereas functional analysis of MAAP partially sheds light on virus egress, genome packaging, and the improvement of vector production. However, functional inconsistencies among serotypes limit the determination of the absolute roles that AAP or MAAP have during the virus life cycle to enhance our knowledge in basic AAV biology and to potentially improve manufacturing processes. Modified MAAPs have shown the capability to boost virus release from cells, and their engineering presents a promising avenue to simplify rAAV production by only processing the media of vector-producing cells. In addition, MAAP ablation or selected MAAP mutants can boost total virus yield, potentially allowing the generation of high vector yields. However, further investigations are necessary to utilize/engineer these small nonstructural proteins in rAAV production. In this context, the identification of cellular or viral factors interacting with AAP and MAAP is crucial for ascribing their specific functions in the virus life cycle. This understanding not only aids in clarifying their roles but also paves the way for defining innovative approaches in rAAV production.

## Figures and Tables

**Figure 1 viruses-16-01215-f001:**
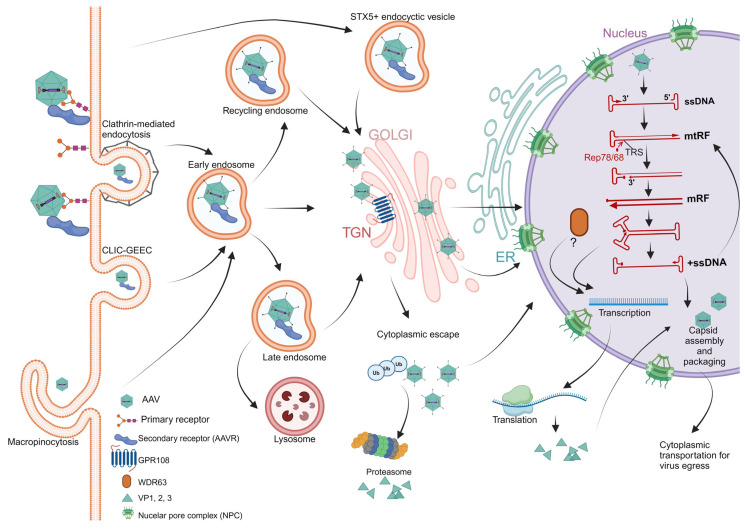
AAV life cycle. AAV is attached to the host cell via glycans. Internalization occurs by various endocytic pathways (clathrin-mediated endocytosis, CLIC/GEEC, or micropinocytosis). Internalized AAV is endocytosed in the early endosome, where the externalization of VP1u occurs. VP1u-externalized AAV then traffics towards the nucleus by following a path through the TGN, Golgi, and ER, or after cytoplasmic escape. Escaped AAV can be degraded after ubiquitination in the proteasome. Entry into the nucleus occurs via nuclear pore complexes (NPC). Once AAV is localized into the nucleus, the capsid disassembles, and viral proteins are expressed following transcription. Viral genome replication follows a rolling-hairpin DNA replication (RHR) model [47,48]. The ssDNA genome is extended from the 3′-OH of the ITR to form a monomer turnround replicative form (mtRF). The hairpin is nicked at a terminal resolution site (TRS) by Rep78/68. The hairpinned end is unwound, and the 3′-end formed by Rep cleavage is extended to the end of the template strand. The ends of each strand refold into their alternative self-base pairing hairpin structures, and the full-length DNA synthesis from the 3′-primer at the left end of the genome produces one ssDNA genome and one mtRF DNA, which can each serve as a substrate for an additional round of replication. WDR63 may play a critical role in rAAV genome transcription [41]. Assembly of progeny virion occurs in the nucleus. Progeny virions then exit from the nucleus and are trafficked to the cytoplasm to be released. Created in Biorender.com (accessed on 25 July 2024).

**Figure 2 viruses-16-01215-f002:**
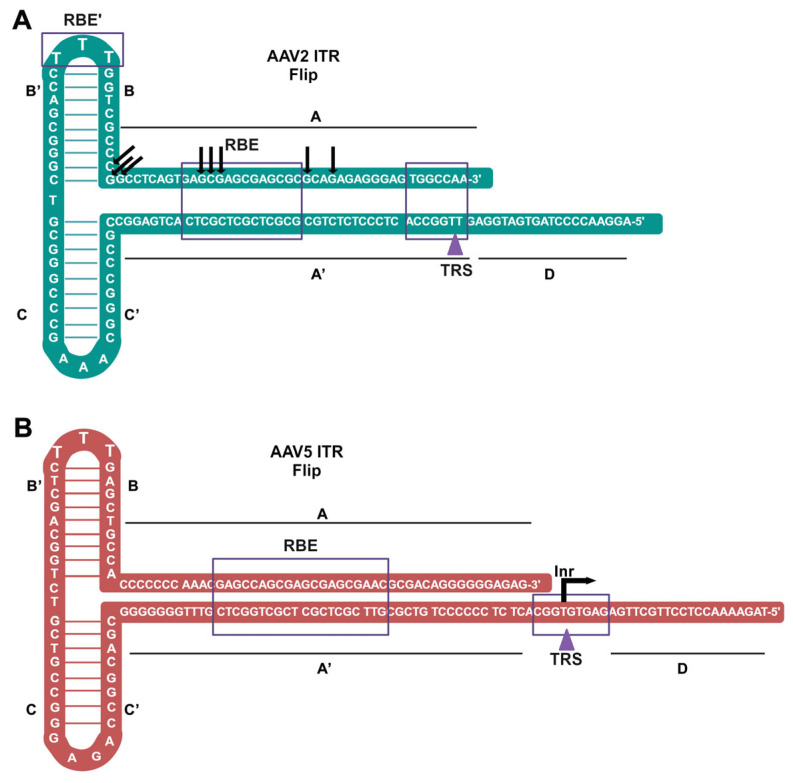
The ITR sequences and structures of AAV2 (**A**) and AAV5 (**B**). Both AAV2 and AAV5 ITRs are diagrammed in a “flip” orientation. The complementary A/A’, B/B’, and C/C’ sequences are labeled. The D-sequence is contiguous with the ssDNA genome body. The terminal resolution site (TRS) consists of the cleavage site (purple arrow) and the flanking nucleotides in the purple line box. Both the AAV2 and AAV5 Rep-binding elements (RBEs) consist of >(GAGC)3, as shown in the purple-lined box. In the ITRs of AAV1–4, 6, and 7 ITRs, multiple transcription start sites (TSSs), indicated by arrows, were found to cluster at the RBE [57]. The Inr of the AAV5 ITR has one TSS at nt 142 [58]. The AAV5 ITR is only 58% homologous with the AAV2 ITR.

**Figure 4 viruses-16-01215-f004:**
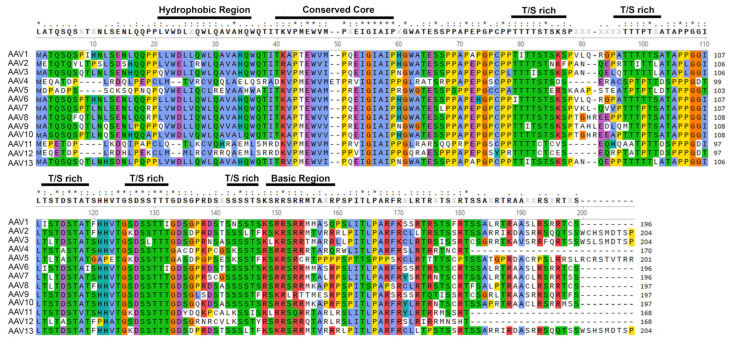
Alignment of AAP sequences of AAV1–13. Predicted AAP protein sequences derived from +1 frame-shifted ORF in the *Cap* genes of AAV1–13 are aligned using MUSCLE and colored with the Clustal X scheme. Consensus amino acids are shown with a threshold of 50%. The CUG codon is conserved as the initiator of translation for all AAPs. Predicted AAP protein sequences are obtained from corresponding cap gene sequences as follows: AAV1-AF063497, AAV2-AF043303, AAV3B-AF028705, AAV4-NC_001829, AAV5-AF085716, AAV6-AF028704, AAV7-NC_006260, AAV8-AF513852, AAV9-AY530579, AAV10-AY631965, AAV11-AY631966, AAV12-DQ813647, and AAV13-EU285562. Domains or motifs of predicted amino acids are underlined. Threonine/serine, T/S, rich. Various motifs are predicted and indicated. Amino acids identical in all 13 proteins are indicated by (*), strongly similar amino acids by (:), and weakly similar by “(.)”.

**Figure 5 viruses-16-01215-f005:**
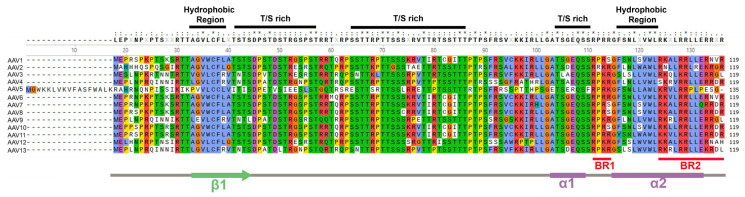
Alignment of MAAP sequences of AAV1–13. Predicted MAAP protein sequences derived from +1 frame-shifted ORF in the *Cap* gene are aligned using MUSCLE and colored with the Clustal X scheme. Consensus amino acids are shown with a threshold of 50%. Predicted MAAP protein sequences are obtained from corresponding *Cap* gene sequences as follows: AAV1-AF063497, AAV2-AF043303, AAV3B-AF028705, AAV4-NC_001829, AAV5-AF085716, AAV6-AF028704, AAV7-NC_006260, AAV8-AF513852, AAV9-AY530579, AAV10-AY631965, AAV11-AY631966, AAV12-DQ813647, and AAV13-EU285562. Domains or motifs of predicted amino acids are underlined. Amino acids identical in all 13 proteins are indicated by (*), strongly similar amino acids by (:), and weakly similar by “(.)”.

**Figure 6 viruses-16-01215-f006:**
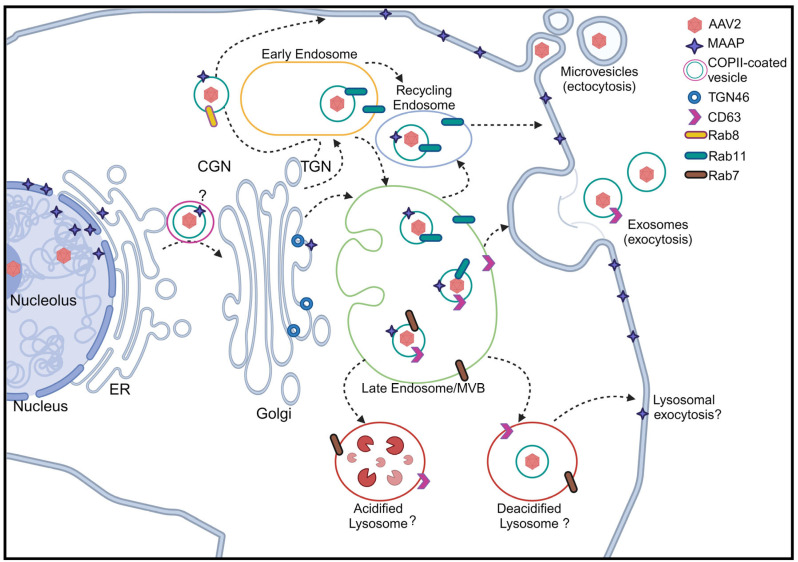
Putative egress routes for AAV2 in nonlytic infection. Assembled AAV2 virions are exported from the nucleus to the endoplasmic reticulum (ER), and then reach the cis-Golgi Network (CGN) to the *trans*-Golgi Network (TGN), which may be mediated by COPII-coated vesicles, like during MVM egress [107]. In one way, virions are packaged in Rab8-positive cargos, which are directed to the plasma membrane for release as microvesicles. On the other hand, virions-containing vesicles can reach early endosomes marked with Rab11 and then traffic to recycling endosomes or late endosomes. The virions in the recycling endosomes can be released as microvesicles. The virions in late endosomes marked with Rab11 and CD63 can be released as exosomes. Rab7-marked late endosomes are led to lysosomes, which will later be degraded in acidified lysosomes or potentially released in deacidified lysosomes by lysosomal exocytosis. MAAP, as marked with stars, could be involved in each trafficking pathway from the nucleus to the plasma membrane as marked. MVB, multivesicular body. Created with BioRender.com (accessed on 25 July 2024).

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
