# Peer review of "The Expression and Function of the Small Nonstructural Proteins of Adeno-Associated Viruses (AAVs)"

_viruses, 2024, doi:10.3390/v16081215_

Round 1

Reviewer 1 Report

Comments and Suggestions for Authors

1. The author gave a nice and clear introduction on AAV basic biology with the emphasis regarding the role of certain non-viral structure proteins, represented by the AAPs.

2. Although the content summarizing the AAV genome structure and life cycle are absolutely required and useful, the text seemed to be a little misbalanced, with less percentage of the description of AAPs (as major forms of non-structual proteins discussed), which appearred until later part of the manuscript. To better serve the title of the review, some rearrange of the text is adviced.

3. From the illustration figures, especially as in Figure6, the representation regarding the role of AAP (and other non-structual proteins) may be more emphasized for easier identification.

4. In section 5.3 or 4.2.3, the introduction to protein X, may need more elabration, for examples on the protein generic features or variations.

5. The listing of bibliography needs to match strictly the required format of the submitted journal.

Reviewer 2 Report

Comments and Suggestions for Authors

The article "The Expression and Function of the Small Nonstructural Proteins of Adeno-Associated Viruses (AAVs)" written by Kuz et al. discusses the function of the recently identified AAV small, non-structural proteins and emphasizes their potential in recombinant AAV vector engineering. The article is well-written and aligns well with the scope of the journal and the interests of its audience.

Major Comments:

1. The first four pages of the article provide valuable but very general information on AAV trafficking and ITR structure. This section is lengthy and not central to the current review. Moreover, it appears to be independent and does not contribute to the understanding of the rest of the article. I suggest either removing this part entirely, as the subsequent chapter on genome organization serves as a suitable entry point, or, if the authors wish to retain the trafficking discussion due to its later relevance to egress, they should focus on the viral components (Adeno/AAV) within this context. This may necessitate moving the figure to section 5 to prevent redundancy. The contribution of most AAV proteins, especially the small non-structural ones, is likely more relevant after nuclear entry. This section can be expanded in the figure to illustrate the involved processes. Mentioning recent observations of cap protein involvement in mRNA expression could also be a valuable addition.

2. The section on AAV genome organization (Section 4) is appropriate for the context of the article. However, the separate figure for AAV5 seems redundant; a mention in the text would suffice.

3. In Section 4.2.2, the discussion on MAAP begins very abruptly, unlike the section on AAP. It would improve readability if the MAAP section starts with the discovery of MAAP, followed by a discussion of the protein's functional domains.

4. The authors discuss AAP's ability to trigger VP3-only particle formation (lines 297-301) and reference three studies (98-100). While reference 99 (Early et al.) employed VP3-only particles, references 98 (Große et al.) and 100 (Maurer et al.; partly) used full VP1-3 particles.

Minor Comments:

1. Gene names should be in italics, including in figure legends.
2. Regarding VP ratios, it might be beneficial to incorporate the recent paper challenging the traditional 1:1:10 ratio and demonstrating a heterogeneous population (https://www.nature.com/articles/s41467-021-21935-5).
3. Since directed evolution is discussed in the context of MAAP engineering for vector production, it might also be valuable to include this topic for AAP (https://pubmed.ncbi.nlm.nih.gov/29978729/).
4. Line 360: Please specify the AAV serotype.
5. Line 444: AAP has not been significantly linked to "packaging efficiency," i.e., the formation of "full" particles or shifting the full-to-empty equilibrium. How do the authors conclude that it stands out as a candidate to enhance such efficiency?
